# Bee Venom Melittin Disintegrates the Respiration of Mitochondria in Healthy Cells and Lymphoblasts, and Induces the Formation of Non-Bilayer Structures in Model Inner Mitochondrial Membranes

**DOI:** 10.3390/ijms222011122

**Published:** 2021-10-15

**Authors:** Edward Gasanoff, Yipeng Liu, Feng Li, Paul Hanlon, Győző Garab

**Affiliations:** 1STEM Program, Science Department, Chaoyang KaiWen Academy, Beijing 100020, China; 1704080008@cy.kaiwenacademy.cn (Y.L.); lifeng85315@126.com (F.L.); mrbiol1963@gmail.com (P.H.); 2Belozersky Institute for Physico-Chemical Biology, Lomonosov Moscow State University, 119991 Moscow, Russia; 3Department of Physics, Faculty of Science, University of Ostrava, 710 00 Ostrava, Czech Republic; 4Biological Research Center, Photosynthetic Membranes Group, Institute of Plant Biology, H-6726 Szeged, Hungary

**Keywords:** mitochondrial bioenergetics, melittin, cytotoxicity, respiratory control index, T cell leukemia, inner mitochondrial membranes, EPR, ^31^P-NMR, native PAGE, AutoDock modeling

## Abstract

In this paper, we examined the effects of melittin, a bee venom membrane-active peptide, on mitochondrial respiration and cell viability of healthy human lymphocytes (HHL) and Jurkat cells, as well as on lymphoblasts from acute human T cell leukemia. The viability of melittin-treated cells was related to changes in O_2_ consumption and in the respiratory control index (RCI) of mitochondria isolated from melittin-pretreated cells as well as of mitochondria first isolated from cells and then directly treated with melittin. It was shown that melittin is three times more cytotoxic to Jurkat cells than to HHL, but O_2_ consumption and RCI values of mitochondria from both cell types were equally affected by melittin when melittin was directly added to mitochondria. To elucidate the molecular mechanism of melittin’s cytotoxicity to healthy and cancer cells, the effects of melittin on lipid-packing and on the dynamics in model plasma membranes of healthy and cancer cells, as well as of the inner mitochondrial membrane, were studied by EPR spin probes. The affinity of melittin binding to phosphatidylcholine, phosphatidylserine, phosphatidic acid and cardiolipin, and binding sites of phospholipids on the surface of melittin were studied by ^31^P-NMR, native PAGE and AutoDock modeling. It is suggested that the melittin-induced decline of mitochondrial bioenergetics contributes primarily to cell death; the higher cytotoxicity of melittin to cancer cells is attributed to its increased permeability through the plasma membrane.

## 1. Introduction

Studies on mitochondrial bioenergetics in different pathophysiological conditions not only aim at understanding the changes in molecular mechanisms of energy-converting membranes that lead to pathology, but also aim at developing novel pharmaceuticals capable of mitigating and/or reversing the pathological changes. The molecular changes in mitochondrial bioenergetics leading to pathophysiological conditions, such as cardiovascular disease, neurodegeneration, inflammation and aging, are being extensively investigated. However, molecular changes in mitochondrial bioenergetics, particularly related to changes in structure and dynamics in the inner mitochondrial membrane, that lead to cancer have not yet been widely studied.

Membrane-active peptides of cationic nature, such as synthetic Szeto–Schiller tetrapeptides and cobra venom polypeptides, have been successfully used in probing the molecular mechanisms of mitochondrial bioenergetics in states of health and dysfunction [1,2]. Molecular details of cardiolipin-targeting by Szeto–Schiller tetrapeptides that trigger a variety of physiological reactions, leading to the rejuvenation of important mitochondrial activities in dysfunctional and aging mitochondria, are being extensively investigated [1]. Application of cobra venom polypeptides in probing the structure-function mechanisms of mitochondrial membranes led to new understandings about the role of non-bilayer lipid structures in mitochondrial bioenergetics [2].

Bee venom melittin is a 26 amino-acid residues cationic membrane-active peptide that is made of six basic, five polar and 15 hydrophobic residues [3]. Melittin exhibits an array of pharmacological effects that include anti-inflammatory [4], anti-arthritic [5], antimicrobial [6], and anticancer [7,8,9,10,11,12] activities. Melittin targets plasma cell membranes [13] and disturbs the tight bilayer packing of lipids, which modulates activities of membrane-associated enzymes [14]. The exact molecular mechanism(s) related to the modulation of membrane dynamics and phospholipid-packing by melittin, particularly associated with the anti-cancer activity of melittin, is not well understood. 

The outer surface of the plasma membranes of healthy cells is made predominantly of phosphatidylcholine (PC) [15]. The polar head of PC includes cationic choline and anionic phosphate groups making PC electrically neutral. However, it is the choline group of PC that is exposed on the outer surface of plasma membranes, making the outer surface of healthy cells act as a positively charged shields that repel basic protein toxins away from the cell surface. Melittin is believed to be the only cationic natural protein toxin that can breach the cationic shield on the membrane surface of healthy cells [16]. Melittin has a 3D structure of a slightly bended alpha-helix [17]. The side chains of hydrophobic amino acid residues are exposed on the inside of the helix bend, while the side chains of charged and polar residues are exposed on the outside of the helix bend [17]. Melittin approaches the surface of PC membranes with the long axis of the helix parallel to the plane of the bilayer [17]. The side chains of hydrophobic residues oriented toward the membrane surface help to overcome electrostatic repulsion between the basic residues of melittin and the choline shield on the membrane surface. Once the hydrophobic side chains of melittin plunge into the PC membrane and reach the alkyl chains of PC, the basic residues of melittin bind electrostatically to phosphate groups of PC molecules. This electrostatic attraction keeps melittin at the interface between the polar region of lipid heads and the non-polar region of alkyl chains, and does not allow melittin to penetrate the inner membrane monolayer [17,18]. Localization of melittin only at the outer monolayer increases the surface area of the outer monolayer over the surface area of the inner monolayer, which creates asymmetric interfacial area tension, leading to a disturbance in the bilayer-packing of lipids [19,20]. 

Acidic phospholipids, which are normally present on the inner monolayer of plasma membranes in healthy cells, accumulate in the outer monolayer of plasma membranes in cancer cells [21,22]. Due to the electrostatic attraction of melittin to acidic phosphatidylserine (PS), which has been shown in our recent study in lipid dispersions and unilamellar liposomes [20], it has been suggested that melittin may interact with membranes of cancer cells with greater avidity than with membranes of healthy cells [20]. Although we have recently examined the molecular binding of melittin to the PS-containing membrane in model lipid systems [20], the interaction of melittin with membranes containing other acidic phospholipids such as phosphatidic acid (PA), another phospholipid found on the outer leaflet of cancer cells [21,22], and cardiolipin (CL), a signature phospholipid of IMM [2], have not yet been investigated. Molecular mechanisms leading to melittin-induced disintegration of PS, PA, or CL-containing membranes have never been studied before.

In this paper, we present our studies on melittin cytotoxicity and its effects on mitochondrial respiration and coupling of IMM in samples of healthy human lymphocytes and lymphoblasts derived from acute T cell leukemia, along with our EPR studies on the effects of melittin on the dynamics and phospholipid-packing in three model membrane systems made of: (1) pure PC, which served as a model of healthy plasma cell membranes; (2) PC containing either 20 mol% PS or 20 mol% PA, which served as a model of cancer plasma cell membranes; and (3) PC containing 20 mol% CL, which served as a model of the inner mitochondrial membranes. We also present in this paper our ^31^P-NMR and native PAGE findings on the avidity of melittin binding to PC, PS, PA and CL, along with the AutoDock simulation of phospholipids binding to the molecular surface of melittin. Overall, the results of this study demonstrate that melittin is three times more cytotoxic to lymphoblastic leukemia cells than to healthy human lymphocytes. Although melittin reduces the mitochondrial respiration and the coupling of inner mitochondrial membranes (IMM) with equal efficiency both in lymphoblasts and healthy lymphocytes, the higher concentration of melittin in the cytosol of lymphoblasts, which are driven by the higher permeability of melittin through plasma membranes of lymphoblasts, contributes to the higher diminishment of the mitochondrial respiration and IMM coupling in lymphoblasts than that in healthy lymphocytes. Finally, we show, for the first time, that melittin targets CL and forms non-bilayer structures in CL-containing membranes. This finding allows us to suggest that melittin at very low concentrations may act in an opposite way to melittin action at higher concentrations. We suggest that melittin at very low concentrations may act similarly to the CL-targeting SS-tetrapeptides that rejuvenate mitochondrial functions and remodel mitochondrial membranes, leading to tissue regeneration during aging and pathophysiological conditions [1,2].

## 2. Results

### 2.1. Melittin Cytotoxicity on Lymphoblasts and Healthy Lymphocytes

The cytolytic activity of melittin and its effects on the oxygen consumption of human leukemia cells and healthy human lymphocytes have been reported by Mirtalipov three decades ago [23]. For leukemia cell samples, Mirtalipov chose T lymphoblastic leukemia cells and B lymphoblastic leukemia cells. Normal peripheral blood lymphocytes, normal T cells and normal B cells were chosen as samples of healthy lymphocytes. There was no difference observed in the melittin effects on cytolysis and respiration between two samples of leukemia cells and between three samples of healthy lymphocytes; however, melittin’s effects on cytolysis and respiration of leukemia cells were three times stronger than on healthy cells [23]. Mirtalipov suggested that mitochondria in cancer cells are more susceptible to attack by melittin than mitochondria in healthy cells [23]. In this paper, we extended Mirtalipov’s study and examined the effects of melittin on cytolysis and respiration of cancer and healthy cells via studying its effect on the structure and dynamics of model membranes mimicking the plasma membrane of cancer and healthy cells, and of the IMM. Since there was no difference reported in the melittin effects between two cancer cell types and three healthy cell types, we have chose only one cancer cell type, Jurkat cells (lymphoblasts derived from human acute T cells leukemia), and one healthy cell type, normal human peripheral blood lymphocytes.

As can be seen in Table 1, the cytotoxic activity of melittin was significantly higher on Jurkat cells than on healthy lymphocytes. At the concentration of 10^−4^ M melittin, the cytotoxicity was about three times higher to Jurkat cells than to healthy lymphocytes. Mellitin at 10^−4^ M killed about 27.8% of healthy human lymphocytes and about 78.5% of Jurkat cells. Interestingly, there was a negligible increase in the cytotoxicity of melittin up to a melittin concentration of 10^−6^ M, after which there was a sharp increase in its cytotoxicity (Table 1). This may suggest that a melittin concentration of 10^−5^ M represents a threshold at which the viability of healthy human lymphocytes can no longer be properly sustained. Contrary to the mode of melittin cytotoxicity on healthy lymphocytes, melittin cytotoxicity on Jurkat cells was gradually increasing with the increase in the concentration of melittin (Table 1). These observations may suggest that there are different mechanisms that could possibly exist beyond the cytotoxic activity of melittin in samples of healthy lymphocytes and Jurkat cells.

### 2.2. Melittin Affects Respiration of Lymphoblasts and Healthy Lymphocytes

Melittin is a membrane-active peptide that binds exclusively to lipids but not to proteins of biological membranes. We have recently shown that melittin synergizes the lipid hydrolytic activity of water-soluble phospholipase A_2_ (PLA_2_) not through the direct binding to the enzyme but by affecting the lipid substrate interface, making it more conducive for PLA_2_ enzymatic activity [19]. There are no reports about melittin affecting mitochondrial energetics through the direct binding to proteins of the electron transport chain, ATP synthase, or enzymes of the Krebs cycle. Thus, melittin affects mitochondrial respiration through its interaction with the lipid phase of biological membranes. To examine whether melittin affects mitochondrial respiration through its interaction with plasma cell membranes or through its direct interaction with mitochondrial membranes, we carried out two sets of separate experiments. In the first set of experiments, we estimated the respiratory control index (RCI) of mitochondria isolated from healthy human lymphocytes (HHL) and from Jurkat cells which were pretreated with melittin; i.e., melittin was allowed to interact with the cells’ plasma membranes. In another set of experiments, we directly treated mitochondria isolated from HHL and Jurkat cells with melittin and then estimated the RCI values of treated and untreated (control) mitochondria. We estimated RCI by measuring the mitochondrial oxygen consumption in state 3 and dividing it by the oxygen consumption in state 4. State 3 is defined as a state with a low ATP/ADP ratio and a high oxygen consumption, while state 4 is defined as a state with a high ATP/ADP ratio (low ADP) and low oxygen consumption corresponding mainly to proton leak. High RCI values correspond to structurally intact and highly coupled IMM, while low RCI values correspond to structurally broken and poorly coupled IMM with proton leaks.

The control mitochondrial samples isolated from HHL and Jurkat cells untreated with melittin had high RCI values (6.5 for HHL and 6.8 for Jurkat cells), consistent with the highly coupled and functionally active mitochondria (Table 2). When HHL and Jurkat cells were treated with 10^−5^ M melittin, the RCI value for mitochondria isolated from HHL was 4.5, which is 69.2% of the RCI of control mitochondria from HHL, while the RCI value for mitochondria from melittin-treated Jurkat cells was 1.7 which is 25.0% of the RCI of control mitochondria from untreated Jurkat cells (Table 2). This observation supports the suggestion of Mirtalipov that mitochondria in cancer cells are more susceptible to the action of melittin than mitochondria in healthy cells [23]. However, when mitochondria isolated from HHL and Jurkat cells were subjected to the direct treatment by melittin, the RCI values of mitochondria for HHL and Jurkat cells were, respectively, 0.95 and 0.94, which are only 18.6% and 17.9% of the RCI values of untreated mitochondria from HHL and Jurkat cells (Table 2). Such a low level of RCI values is consistent with the highly uncoupled mitochondria. It should be noted that very low values of oxygen consumption in state 3 and state 4 were virtually identical in the directly treated samples with melittin mitochondria both from HHL and Jurkat cells (Table 2). This is an unusual observation as oxygen consumption in state 3 should be higher than that in state 4 in structurally intact and functional mitochondria. This unusual observation suggests that direct treatment of mitochondria by melittin severely disintegrates the structural integrity of IMM. It should also be noted that the RCI values of mitochondria from HHL and Jurkat cells directly treated with melittin were not only low but also closely identical, specifically 0.95 and 0.94, suggesting that mitochondria of healthy human lymphocytes and human leukemia cells are equally susceptible to the direct attack by melittin. Thus, the above set of data allows one to suggest that the differences in the effect of melittin on healthy human lymphocytes and T leukemia cells are modulated not by the differences in the susceptibility of mitochondria of healthy cells and cancer cells to the melittin action, but by the differences in the ability of melittin to affect the structure of plasma membranes in healthy lymphocytes and in Jurkat cells; in turn, this generates different consequences related to mitochondrial respiration in cancer and healthy cells.

### 2.3. EPR Study on Melittin Interaction with Model Membranes

To assess the differences in melittin action on the structure and dynamics of plasma membranes in healthy lymphocytes and Jurkat cells, and that of the inner mitochondrial membrane, we studied how melittin affects the order of bilayer-packing and the rotational movement of phospholipids in model membranes with the phospholipid composition that resembles that of the plasma membrane outer leaflet in both normal human peripheral blood lymphocytes (HPBL) and in lymphoblasts derived from acute T cell leukemia (Jurkat cells), and with a composition resembling that of the IMM of HPBL and Jurkat cells. It has been determined that PC is a major neutral phospholipid and that both PS and PA are the major anionic phospholipids in the plasma membrane of human white blood cells [24]. It has also been determined that the outer leaflet of plasma membranes of HPBL is mostly made of PC, but the outer leaflet of the plasma membrane of Jurkat cells includes, in addition to PC, 10 mol% of PS and 10 mol% of PA (which is the same in B-acute lymphoblastic leukemia) [24]. The phospholipid composition of IMM in HPBL and in Jurkat cells were determined to be same, including 40 mol% PC, 35 mol% phosphatidylethanolamine (PE), 20 mol% CL, 3 mol% phosphatidylinositol (PI) and 2 mol% PS [24]. A model membrane system in this study was a lipid film of oriented multibilayers including 5-doxylstearic acid (5-DSA), an EPR spin probe which is very sensitive to changes in the packing order and rotational movement of lipids [25]. In oriented multibilayer films, anionic lipids are distributed equally on both sides of a bilayer; therefore, a multibilayer film containing 60 mol% PC, 20 mol% PS and 20 mol% PA closely resembles the phospholipid composition of the outer leaflet of plasma membranes of Jurkat cells. However, a multibilayer film of the above phospholipid composition had a low level of phospholipids’ packing order, which did not allow for an accurate calculation of the EPR parameters. The same problem occurred with oriented multibilayer films that resembled the phospholipid composition of IMM in HPBL and in Jurkat cells. Therefore, we prepared two separately oriented multibilayer films, with each containing 80 mol% PC and either 20 mol% PS or 20 mol% PA; both of each proved to be efficient for the accurate calculation of 5-DSA EPR parameters. These two membrane systems allowed us to separately model the interaction of melittin with the two important anionic phospholipids, namely PS and PA, on the outer leaflet of Jurkat cells’ plasma membranes [24]. Regarding the model of IMM, we prepared oriented multibilayer films made of 80 mol% PC and 20 mol% CL, which also proved to be suitable for the accurate calculation of 5-DSA EPR parameters. This lipid film allowed us to model the interaction of melittin with CL, a signature phospholipid of mitochondria and a key phospholipid in regulating bioenergetics in the IMM [2].

The lipid-packing in bilayers is characterized by a strong anisotropic molecular orientation of phospholipids. The 5-DSA spin probe is a stearic acid with the free radical group attached to the fifth carbon atom from the carboxyl group. In the multibilayer lipid films, the long molecular axis of 5-DSA aligned along the long axes of fatty acid chains of lipids. Thus, 5-DSA incorporated in the multibilayer film showed strong molecular anisotropy. In the highly ordered multibilayer lipid films not containing melittin, 5-DSA showed strong EPR spectral anisotropy that featured the presence of a wide resonance line when the magnetic field was parallel to the bilayer normal, as well as a narrow resonance line when the magnetic field was perpendicular to the bilayer normal (Figure 1). With the increase in melittin concentration in the PC + 20 mol% CL film, the spectral anisotropy of 5-DSA progressively decreased, which is seen from the broadening of a resonance line when the bilayer normal was perpendicular to the magnetic field (Figure 1A). A virtually complete overlap of the EPR spectral lines obtained at parallel and perpendicular orientations of a lipid film in the magnetic field was observed when the melittin to lipid molar ratio reached 0.015 (Figure 1A), which is consistent with the change of bilayer phospholipid-packing to non-bilayer-packing [25]. In the PC + 20 mol% PS membrane, melittin induced a modest broadening and overlap of EPR spectral lines recorded when the magnetic field was applied perpendicular and parallel to the membrane normal (Figure 1B). Although changes in the EPR spectra in the PC + 20 mol% PS film clearly reflected the membrane-disturbing effect of melittin, the spectral anisotropy of 5-DSA remained unchanged even when the melittin to lipid molar ratio reached 0.015 (Figure 1B). Similar changes in ERP spectra from the differently oriented lipid films in the magnetic field were obtained in the PC + 20 mol% PA membrane and in the pure PC membrane treated with melittin. It should be noted that the membrane-disturbing effect of melittin was less obvious in the pure PC membrane than that in the PC + 20 mol% PA membrane (spectra not shown).

To quantitively analyze the effect of melittin on phospholipid-packing in oriented bilayer membranes, we calculated the B/C ratio from the EPR spectra of melittin-treated and untreated membrane films. The B/C ratio is very sensitive to macroscopic disordering [25,26,27] and to the formation of non-bilayer structures in a lipid phase [25,26,27]. In agreement with the EPR spectra, melittin induced a significant decrease in the B/C ratios in PC+ 20 mol% CL membranes and a smaller decrease in PC+ 20 mol% PS or 20 mol% PA membranes (Figure 2). The (B/C)_I_/(B/C)_0_ values from 0.8 to 0.6 indicate that the lipid membranes experienced local disturbances, causing local changes in bilayer lipid-packing, while the (B/C)_I_/(B/C)_0_ values below 0.5 indicate that the non-bilayer lipid-packing became the dominant lipid phase of the membrane [25,26]. Melittin exerted an insignificant effect on pure PC membranes in the range of a melittin:lipid molar ratio up to 0.009. However, further increase in melittin concentration produces a sharp decline in the bilayer packing order, as evidenced by the decline in (B/C)_I_/(B/C)_0_ values from 0.99 to 0.80 (Figure 2). It has been postulated and supported by experimental evidence that melittin binds to the outer monolayer of PC membranes with a long melittin axis parallel to the bilayer plane [16,17]. Melittin does not penetrate the inner monolayer of PC membranes, which causes an increase in the surface area of outer monolayers faster than that in the inner monolayers [16,17,18]. To release the stress from the asymmetric change in monolayer surface areas, melittin at the melittin:lipid molar ratio of 1:100 changes the orientation of its long molecular axis, from parallel to perpendicular, to the bilayer plane, which increases the structural disturbance in the membrane and facilitates the formation of membrane pores [16,17,18]. A sharp decline in the (B/C)_I_/(B/C)_0_ values with melittin:lipid molar ratios between 0.009 and 0.011 in PC-oriented films (Figure 2) most likely represents an increase in the disturbance of bilayer-packing caused by melittin when it changes the orientation of its long molecular axis from parallel to perpendicular, to the membrane plane.

Another parameter which we calculated from the EPR spectra of melittin-treated and untreated membrane films was the *S* parameter, which is sensitive to changes in the rotational movement of spin probes that mimic the dynamic behavior of lipids in membranes [25,26,27]. It should be noted that acidic 5-DSA mimics the behavior of acidic phospholipids in our study. As shown in Figure 3, an increase in melittin concentration caused an increase in the values of the *S* parameter, which reflects a decrease in the rotational movement of 5-DAS in the lipid membranes. This was caused by the binding of 5-DSA to melittin, which was driven by the electrostatic attraction of acidic 5-DSA to basic melittin. 5-DSA competes with acidic phospholipids for binding to melittin. In pure PC membranes in the absence of acidic phospholipids, there are no competitors for the 5-DSA binding to melittin. This explains the higher values for the *S* parameter found in pure PC membranes than that in PC membranes containing acidic phospholipids (Figure 3). In the PC + 20 mol% CL membrane, 5-DSA of molecular mass 385 Da shall outcompete CL of molecular mass 1422 Da for binding to melittin. This notion agrees with the data in Figure 3, which demonstrate that the values of the *S* parameter in CL-containing melittin-treated membrane are just slightly lower than in pure PC membrane treated with melittin. A molecular mass of 5-DSA (385 Da) is not much lighter than that of PS (792 Da) or PA (649 Da). This means that 5-DSA competes with PS or PA to bind to melittin not as efficiently as it does with CL. This notion agrees well with the data in Figure 3, which shows that the increase in the *S* parameter in PS or PA-containing membranes treated with melittin is considerably lower than that in the CL-containing melittin-treated membrane. Overall, from the data on the changes in the B/C ratio and *S* parameter caused by the interaction of melittin with model membranes, one can conclude that melittin immobilizes the rotational movement of acidic phospholipids and triggers the formation of non-bilayer lipid structures in the CL-containing membranes, in which non-bilayer structures become the dominant lipid phase in the membrane at the melittin to lipid molar ratio of 0.011 and above. In the PS and PA-containing membranes, melittin caused local disturbances in the bilayer-packing of phospholipid with the formation of local non-bilayer structures, representing a minor lipid phase in the membrane. Regarding the pure PC membrane, our EPR study agrees well with previous reports [16,17,18,19] indicating that melittin disturbs the bilayer-packing of PC through the mechanism of asymmetric change in surface areas of two monolayers. At melittin to PC molar ratios exceeding 0.01, the formation of locally non-bilayer-packed phospholipids is associated with the creation of membrane pores.

### 2.4. ^31^P-NMR Study on the Melittin Association with Phospholipids in Buffers

It has been recently demonstrated that the cobra-venom cationic protein cardiotoxin CTII tightly binds to CL to form an oligomeric complex, including four cytotoxins and 16 CL molecules [28]. It should be noted that CTII phenocopies the ability of the dicyclohexylcarbodiimide-binding protein, a subunit of F_0_ sector of ATP synthase, to form protein-CL oligomers [28]. The cardiotoxin-CL oligomeric complex remained intact in a Tris-HCl buffer with Triton X-100 [28]. Triton X-100 is a comparatively mild non-denaturing detergent and a common non-ionic surfactant used in biochemical research to solubilize proteins by stripping lipids from the protein surface. The fact that Triton X-100 in a buffer was not able to dissolve the cardiotoxin-CL complex is consistent with a strong association between cardiotoxin and CL. In our present study, we investigated whether CL, PS, PA, or PC are able to form a strong association with melittin by using methodology described previously [28,29], in which phospholipids and melittin are incubated in Tris-HCl buffer with Triton-X 100. In brief, in Tris-HCl buffer, including Triton X-100 and phospholipids at a low concentration, phospholipids dissolve in buffer and exist as single molecules that rapidly exchange within 10^−2^ to 10^−4^ s with tiny micelles of Triton X-100, including two or three phospholipids [28,29]. A fast exchange of phospholipids results in a high-isotropic dynamics that yield a narrow ^31^P NMR signal [28,29]. A protein that strongly associates with phospholipids restricts the movement of phospholipids, which greatly broadens the ^31^P NMR signal from phospholipids strongly bound to a protein, which in turn renders restricted phospholipids “invisible” to NMR. The disruption of a protein–phospholipid association by treatment with SDS liberates the restricted phospholipids and restores a narrow ^31^P NMR signal. Following this concept, we tested multiple samples in which melittin was separately mixed with CL, PS, PA or PC in 1% Triton X-100-containing buffer (10 mM Tris-HCl, pH 7.5, 0.5 mM EDTA, 1% Triton X-100) at various melittin to phospholipid molar ratios. As can be seen in Figure 4a, CL dissolved in 1% Triton X-100-containing buffer at 4.5 × 10^−6^ M concentration produced a single narrow ^31^P NMR signal, suggesting that CL rapidly exchanges within tiny micelles made of a few molecules of Triton X-100 and CL. Incubating melittin in Triton X-100 buffer with CL at the melittin to CL molar ratio 1:3 rendered the ^31^P NMR signal from CL invisible (Figure 4b), suggesting that one mole of melittin tightly immobilizes three moles of CL. Adding 1% SDS in this buffer restored the ^31^P NMR signal (Figure 4c), indicating that SDS disrupts a tight melittin–CL association and restores a rapid isotropic mobility of CL. When PS was dissolved at 8.4 × 10^−6^ M concentration in Triton X-100-containing buffer, a single narrow ^31^P NMR signal was produced (Figure 4d), suggesting that PS dissolved in buffer is rapidly exchanging with tiny micelles made of Triton X-100 and PS. It should be noted that the ^31^P-NMR signal from PS (Figure 4d) is located on the high-field side of the spectrum compared to the spectrum of CL (Figure 4a,d), which is consistent with different physico-chemical environments of phosphate groups in PS and CL. Incubating melittin at 2.8 × 10^−6^ M concentration with a PS of 8.4 × 10^−6^ M concentration (i.e., melittin:PS molar ratio 1:3) in 1% Triton X-100-containing buffer did not affect the ^31^P NMR spectrum of PS (Figure 4e). Thus, melittin did not render the ^31^P NMR signal of PS invisible. This strongly suggests that melittin does not immobilize PS in Triton X-100 solution. Incubating melittin with different concentrations of PS in 1% Triton X-100-containing buffer also did not render the ^31^P NMR signal from PS invisible (data not shown). Similar results were obtained when melittin was separately incubated in Triton X-100-containing buffer with PA or PC at different molar ratios (data not shown). Overall, this set of data strongly suggest that melittin is capable of binding strongly only to CL in the presence of Triton X-100, but not to PS, PA or PC.

### 2.5. Oligomerization of Melittin with Phospholipids: Study by Native PAGE 

To estimate the molecular weight of the melittin–CL complex formed after incubating melittin with CL in 10 mM Tris-HCl, pH 7.5, 0.5 mM EDTA, 1% Triton X-100 buffer for one hour at 37 °C, we employed the native PAGE protocol as previously described [30]. We observed a band slightly above 25 kDa, representing the melittin–CL complex (Figure 5 lane 5). Considering that the melittin (2.85 kDa) to CL (1.422 kDa) molar ratio was 1:3, the band slightly above 25 kDa is consistent with a molecular weight of the complex comprised of four melittins (11.4 kDa) and 12 CL molecules (17.1 kDa), totaling to about 28.5 kDa. Protein bands in lanes 2, 3 and 4 in Figure 5 are consistent with monomeric or dimeric forms of melittin free of phospholipids. Overall, one can conclude that the tight bonding in the melittin–CL complex was strong enough to withstand the treatment with Triton X-100, while the bond between melittin and either PS, PA or PC was not strong enough to withstand the same treatment.

### 2.6. Molecular Docking of Melittin with Phospholipids

In this study, we also modelled the interaction of phospholipid polar heads of CL, PS, PA and PC with the molecular surface of melittin by using the AutoDock program according to the methodology previously described [27,28,31]. We were interested in finding specific binding sites for the phospholipids used in this study on the molecular surface of melittin to elucidate the molecular mechanism(s) that may explain the different effects that melittin caused on pure PC membrane and on PC membrane enriched with CL, PS or PA. In our docking studies, we used the entire molecular surface of melittin since we wanted to simulate a situation in which melittin is entirely submerged into a membrane, making the whole molecular surface of melittin available for interaction with the polar heads of phospholipids. The AutoDock software produced nine docked conformations with the highest energies of binding for each of the four ‘receptor-ligand’ pairs, in which melittin acted in a capacity of a receptor and a phospholipid polar head acted in a capacity of a ligand. As one can see from Table 3, which shows the predicted binding affinities at nine binding sites on melittin’s surface, the binding of the CL polar head to the molecular surface of melittin releases more energy than when polar heads of other phospholipids, namely PS, PA or PC, bind to melittin. This means that the CL polar head binds to the melittin surface with a stronger force than the polar heads of other phospholipids. The charged and polar groups of the CL head, which make ionic, ion–polar, and hydrogen bonds with amino acid residues of melittin in the nine binding sites, are listed in the Appendix A. The nine binding sites of the CL polar head represent five binding locations on the molecular surface of melittin. This means that several binding sites are situated in the same location but they are specified by the AutoDock program as different binding sites when the same amino acid residues bind to different groups of the CL polar head. Overall, there are seven charged and polar residues, namely T11, S18, W19, R24, R22, Q25 and Q26, which bind to the CL polar head groups in the five presumed locations on the molecular surface of melittin. Additionally, in binding site #4 (binding affinity = −3.9 kcal/mol), carbon atoms of the CL polar head interact with six non-polar residues, namely 12G, 13L, 14P, 15A, 16L, and 17I, which presumably strengthen the force of the binding of the CL polar head to the surface of melittin in this location. What makes the CL polar head binding conformations to the melittin surface different from that of other phospholipid polar heads (PS, PA and PC) is that in eight out of the nine binding sites, when one phosphate group (PO_4_^−^) of CL is bound to melittin’s surface, another phosphate group, along with one or two negative poles (δ−) of CL polar groups, is oriented away from the melittin’s surface to electrostatically attract basic residues on the surface of neighboring melittin.

In Figure 6, we propose one of the hypothetical models in which melittin dimerization is mediated by the CL polar head binding conformation to the melittin surface, as predicted by the AutoDock modeling for binding side 8 (Appendix A); here, one phosphate, specifically b-PO_4_^−^, of CL makes three bonds (one ionic and two ion–polar) with R22 and G26 residues of melittin (Appendix A), while another phosphate, specifically a-PO_4_^−^, of CL is oriented away from the melittin surface in the same manner the VI-C = O^δ−^ group of CL is (for the designation of CL polar groups, refer to Appendix A). Figure 6 presents two images of the predicted melittin–CL polar head complex, in which the bottom image is flipped 180 degrees horizontally and then vertically. These spatial orientations of the two complexes provide the two centers of electrostatic attraction involving the -NH_3_ group of K7 of both melittins and the VI-C = O^δ−^ group of both CLs (short-range ion–polar attraction), as well as the -NH_3_ group of K7 of both melittins and the phosphate, a-PO_4_^−^, of both CLs (long-range ionic attraction). Should this type of melittin dimerization mediated by CL take place in vitro, as in in our experiment, in buffer with Triton X-100? The space between two complexes could be filled by eight alkyl chains of two CLs. If a space between the melittin dimers is filled not by two but by four CLs, it will take 12 CLs to make a tetramer, as there would be three spaces between four melittins. This hypothetical arrangement for the tetramerization of melittin mediated by CLs agrees well with the 28.5 kDa melittin–CL complex that includes four melittins and 12 CLs, as determined in our experiments involving Triton X-100 as described above. CL molecules situated between melittin molecules are not easily accessible to Triton X-100, which may explain why Triton X-100 was not able to strip CLs from the molecular surface of melittins under our experimental conditions. It should be noted that the proposed polymerization of melittin mediated by CL may take place in vivo in the inner mitochondrial membranes, which contain 20 mol% CL.

Docking of the phospholipid polar heads of PS, PA or PC with melittin by the AutoDock program predicted ionic, ion–polar, hydrogen, and non-polar bonds, as listed in Appendix A. Amino acid residues of melittin involved in making ionic, ion–polar, and hydrogen bonds with polar head groups of PS were K7, T11, L13, A15, L16, S18, K21, R22, Q25 and Q26. Non-polar residues L13, A15, and L16 participated in making hydrogen and ion–polar bonds, respectively, by using the NH^δ+^ group of the peptide bond, which is designated as NH^δ+^_pb_ in the Appendix A. Overall, nine binding sites predicted for PS generated four different binding locations on the molecular surface of melittin for PS polar head groups. Amino acid residues of melittin involved in making ionic, ion–polar, and hydrogen bonds with polar head groups of PA were L16, S18, W19, K21, R22, G25 and G26 (Appendix A), which generated three spatially distinct binding location on the melittin surface. The ion–polar bond was made with residue L16 via its NH^δ+^_pb_ group (Appendix A). The residues that were involved in making ionic, ion–polar and hydrogen bonds with polar head of PC were G12, L16, S18, W19, K21, R22 and Q26, which generated four spatially distinct binding locations on the molecular surface of melittin. Residues G12 and L16 were involved in making hydrogen and ion–polar bonds, respectively, via their NH^δ+^_pb_ groups (Appendix A). Interestingly, carbon atoms of the PC polar head were involved in non-polar interactions in binding sites 3, 4, 6, 7, and 8 (refer to Appendix A for the melittin residues involved in hydrophobic interactions with the PC polar head). A thorough review of all the binding conformations predicted by AutoDock for PS, PA, and PA polar heads on the molecular surface of melittin did not reveal hypothetical opportunities for phospholipid-mediated polymerization of melittin in membranes containing PS, PA or PC. This finding agrees well with our experiments in this study that showed no melittin oligomerization with PS, PA, or PC in buffers containing Triton X-100.

## 3. Discussion

The state of mitochondrial bioenergetics has become a universal indicator for the state of wellbeing of cells, organs and entire organisms. Decline in the activity of respiratory protein complexes, especially CI and CIV, leads to the decreased production of ATP [32,33,34]. An inadequate supply of ATP impairs the functional activities of essential body organs and tissues such as the heart, brain, kidneys, skeletal musculature, nerve cells and cells of the immune system which demand a large amount of energy to support a proper level of physiological activity [2]. Insufficient levels of mitochondrial bioenergetics obstruct muscle contraction, blood circulation, neurotransmission and the repair of injured cells, which lead to a variety of pathophysiological conditions including cardiovascular disease, ischemia, heart failure, neurodegenerative diseases, inflammation, and immunological diseases, each of which may cause the premature death of cells and an entire organism [2]. A scarce supply of bioenergy also impairs the routine paths of programmed cell death [35], resulting in an increased rate of morbidity and mortality [2]. A decline in mitochondrial bioenergetics is also linked to aging [36], considering that with age, the normal physiological architecture of the inner mitochondrial membranes (IMM) crumbles [1,2], leading to a decrease in the surface area of cristae in the IMM, resulting in decreased activity of respiratory protein complexes [32,33]. Therefore, mitochondrial bioenergetics leading to different pathophysiological conditions attracts broad attention of experts from various fields of molecular science, biochemistry and biophysics. An extension of our knowledge in this area supports the development of new pharmaceuticals capable of impeding or reversing pathophysiological reactions caused by impaired mitochondrial bioenergetics.

The essential role of CL in controlling mitochondrial bioenergetics has been studied for several decades [2]. Recently, the cationic Szeto–Schiller (SS) tetrapeptides, which are known as CL-protective compounds, have been synthesized to probe the molecular mechanisms by which cardiolipin regulates mitochondrial bioenergetics [1]. The SS tetrapeptides penetrate cell membranes and cell barriers with tight junctions, such as the blood–brain barrier [37]. The SS tetrapeptides target CL in the IMM and act as therapeutic agents to restore mitochondrial plasticity [38] as well as mitochondrial bioenergetics [39]. The effects of SS tetrapeptides have been widely studied in cells, tissues and animals to understand molecular mechanism(s) of action(s), in which the SS tetrapeptides rejuvenate mitochondrial bioenergetics and reverse pathophysiological processes [1,39]. It has been recently reported that cobra venom cationic membrane-active polypeptides, called cardiotoxins, can also target mitochondrial CL [27,40]. Cardiotoxins penetrate cell membranes through the mechanism of inverted membrane junctions [31]. At very small concentrations, cardiotoxins induce the formation of controlled amounts of non-bilayer structures in the IMM, which increases mitochondrial ATP production [2,28]. It should be noted that at higher concentrations, cardiotoxins disintegrate the physiological structure of the IMM, leading to stagnation of mitochondrial respiration and ATP production [2,28,31].

Mitochondrial bioenergetics of cancer cells has not been studied widely in terms of tracing the differences between mitochondrial bioenergetics of cancer cells and that of healthy cells, and in terms of relating the differences to changes in the structure and dynamics in the IMM in pathophysiology and health. This is probably because mitochondrial bioenergetics of cancer cells are not much different from mitochondrial bioenergetics of healthy cells at the level phospholipid-packing and mobility in the IMM. In the present study, we showed that the levels of mitochondrial respiration and coupling in healthy human lymphocytes (HHL) and in lymphoblasts derived from acute T cells leukemia (Jurkat cells) were very close, as judged by RCI values: 6.5 for HHL and 6.8 for Jurkat cells (Table 2). We used melittin, a cationic membrane-active peptide, to probe the mitochondrial respiration and coupling, and cell viability of Jurkat cells and compared with HHL. When both types of cells were treated with 10^−5^ M melittin, the RCI of Jurkat cells was 25.0% of control (untreated) Jurkat cells, while the RCI of HHL was 69.2% of control (untreated) HHL (Table 2). Cells treated with 10^−5^ M melittin revealed 73.8% viable HHL of control HHL and 25.2% viable Jurkat cells of control Jurkat cells (Table 1). When one compares RCI and the viability of treated HHL, which are 69.2% and 73.8%, respectively, and RCI and the viability of treated Jurkat cells, which are 25.0% and 25.2%, respectively, one can see that RCI and the viability of treated HHL are higher than that of Jurkat cells and that there is a close match between RCI and the cell viability in the same type of cells. This observation may suggest that cytotoxicity of melittin is linked to the suppression of mitochondrial respiration and that Jurkat cells are more susceptible than HHL to melittin action. However, when mitochondria isolated both from HHL and Jurkat cells were subjected to direct treatment by 10^−5^ M melittin, the RCI values were about 0.95 for both types of cells, which is about 18% of the RCI of control (untreated) mitochondria samples. Such a low RCI value strongly suggests that when melittin directly attacks mitochondria, it almost entirely uncouples the mitochondrial membranes and severely suppresses mitochondrial respiration both in HHL and Jurkat cells. This implies that mitochondria of both types of cells are equally susceptible to direct action of melittin, which is probably because the mitochondrial structure and physiology in both types of cells is very similar, if not identical. Thus, the question concerns why intact Jurkat cells are more susceptible than intact HHL to melittin action.

Melittin is a membrane-active peptide and there is no evidence that melittin directly affects the activities of proteins and other non-lipidic physiologically active compounds in the IMM, in cytosol, in the mitochondrial intermembrane space or in the matrix. This means that the effects of melittin described above on HHL and Jurkat cells are linked to the melittin-triggered changes in the lipid phase of plasma cell membranes and in the mitochondrial membranes. Since melittin induced the same change in the RCI values when directly attacking mitochondria of both types of cells, one can conclude that the different effects of melittin on the RCI values and on the cell viability in samples of intact HHL and Jurkat cells are based on different mechanisms of melittin action on plasma cell membranes of HHL and Jurkat cells. This is reasonable as plasma membranes of HHL and Jurkat cells differ in terms of phospholipid composition.

In this study the pure PC membrane served as a model of the plasma membrane of healthy cells, while the PC membrane containing 20 mol% PS or PA served as a model of the plasma membrane of cancer cells, and the PC membrane containing 20 mol% CL served as a model of the IMM. This study showed that melittin affected the membrane structure to the least extent in a model of the plasma membrane of healthy cells. In this membrane, the lowest (B/C)_I_/(B/C)_0_ value was 0.8, which was reached at the melittin to lipid molar ratio of 0.011 (Figure 2). The (B/C)_I_/(B/C)_0_ value of 0.8 reflects a low disturbance level of the membrane barrier at which only a small percent of cationic peptides can penetrate the membrane [25,41]. At the melittin to lipid molar ratio of 0.011 in membranes containing 20 mol% PS or PA within the model of the plasma membrane of cancer cells, the (B/C)_I_/(B/C)_0_ value was 0.7, which is consistent with a higher level of membrane barrier disturbance [25,41]. At the melittin to lipid molar ratio of 0.021, the (B/C)_I_/(B/C)_0_ value in the healthy cell plasma membrane model was just slightly below 0.8, but that in the cancer cell plasma membrane model was close to 0.6 (Figure 2). This observation strongly suggests that it is easier for melittin to penetrate the plasma membrane of cancer cells than that of healthy cells [25,41]. In the model of IMM, the (B/C)_I_/(B/C)_0_ value dropped to 0.45 at the melittin to lipid molar ratio of 0.011 (Figure 2). The (B/C)_I_/(B/C)_0_ value of 0.45 is consistent with a membrane that is mostly made of non-bilayer lipid structures [25,41]. In our previous work, we have shown that a cationic peptide can easily penetrate a membrane containing large amounts of non-bilayer structures [27,41] and that when the amount of non-bilayer structures in the IMM reaches 40% and above mitochondrial ATP productions stagnate [2,28,31] probably because of a significant uncoupling of mitochondria [2]. This implies that melittin can not only effectively penetrate the mitochondrial membrane but can also inflict a pathophysiological state in mitochondrial bioenergetics at relatively small concentrations.

It should be noted that an apparent discrepancy exists between the (B/C)_I_/(B/C)_0_ values in the model membranes of healthy and cancer cells, and the melittin cytotoxicity on HHL and Jurkat cells that we observed in this study. In our previous studies, it has been shown that when plasma membrane has a (B/C)_I_/(B/C)_0_ value of around 0.8, which was observed in our model of healthy cell plasma membranes treated with melittin, the cells maintain cell viability at 37 °C for 6 h [25,41]. When the (B/C)_I_/(B/C)_0_ value of plasma membranes is at about 0.6, the cells maintain cell viability at 37 °C for 1.5 h [25,41]. We observed (B/C)_I_/(B/C)_0_ values above 0.6 in our model of cancer cell plasma membranes treated with melittin. However, our melittin cytotoxicity tests on HHL and Jurkat cells demonstrated that cells started dying after 30 min of treatment with melittin (73.2% viability for HHL and 21.5% viability for Jurkat cells at 10^−4^ M melittin). This apparent discrepancy could be explained by the strong binding avidity of melittin to CL, which was demonstrated in the present study by ^31^P-NMR and native PAGE in our experiments on molecular associations of melittin with CL in Triton X-100 solution. The results of our study suggest that melittin penetrates the plasma membranes of both HHL and Jurkat cells, and once in the cytosol, melittin penetrates the outer mitochondrial membrane and avidly disturbs the physiological structure of the IMM, leading to the rapid death of HHL and Jurkat cells. Thus, one can conclude that melittin does not likely inflict pathology leading to cell death through the disturbance of plasma cell membranes, but rather melittin does this via disrupting the mitochondrial energetics by inducing the formation of large amounts of non-bilayer structures in the IMM. Our previous work on cobra cardiotoxins’ interaction with the IMM [2,28,31] demonstrated that physiologically present non-bilayer lipids (mostly CLs), supposedly immobilized by proteins of the F_o_ subunit of ATP synthase, facilitate ATP production probably via the formation of non-bilayer compartments, with an increased concentration of H^+^ near the F_o_ subunit proton channel. At very small concentrations, cardiotoxins promote further formation of non-bilayer compartments with immobilized phospholipids to further increase ATP production [2,28,31]. However, at a certain threshold of cardiotoxin concentration, the large amount of non-bilayer structures leads to destabilization of the physiological dynamics and structure of the IMM, and to disruption in ATP production. We believe that the melittin concentrations used in this study are above the ‘physiological’ threshold, leading to the disruption of mitochondrial energetics. It should be noted that melittin penetrates the plasma membrane of Jurkat cells more efficiently than that of HHL and thus melittin accumulates in the cytosol of Jurkat cells at a higher rate than in the cytosol of HHL. The results of this study do not show that mitochondria of Jurkat cells are more susceptible to attack by melittin than mitochondria of HHL. However, because Jurkat cells accumulate melittin in the cytosol at a higher concentration due to the higher permeability of melittin through the plasma membranes of Jurkat cells, melittin at a higher concentration disrupts the mitochondrial bioenergetics of Jurkat cells faster than that of HHL. Should this behavior of melittin be proven true on other types of cancer cells melittin may present an attractive opportunity to be used as a potential anti-cancer drug; and as such, the present work warrants further in vitro and in vivo studies on melittin interactions with CL and the IMM.

The application of AutoDock simulation in this paper allowed us to demonstrate the hypothetical structural opportunities for the CL-facilitated polymerization of melittin, which are supported by the high “plasticity” of CL and its tendency to form non-bilayer structures. The same was not found for PS, PA and PC. We have also shown for the first time in this study that melittin forms non-bilayer structures less efficiently in PS and PA-containing membranes than in CL-containing membranes. It appears that melittin may act as a true CL-targeting peptide in the same way as the SS-tetrapeptides. It is possible that at very low concentrations melittin stimulates, similarly to SS-tetrapeptides, the restoration of mitochondrial plasticity [38] and, similarly to cobra cardiotoxins, may trigger at very low concentrations the formation of physiological amounts of non-bilayer structures to facilitate ATP synthase activity [28]. The present work warrants further in vitro and in vivo studies on the melittin-induced non-bilayer structures in CL-containing membranes and on how the controlled modulation of physiological amounts of non-bilayer structures affects the bioenergetics of the IMM. Further studies may prove or disprove potential applications of melittin not only as an anti-cancer agent but also as a pharmaceutical drug capable of mitigating and/or reversing non-cancerous pathophysiological developments in mitochondrial bioenergetics.

## 4. Materials and Methods

### 4.1. Reagents

Melittin from bee venom, egg yolk L-α-phosphatidylcholine (PC), cardiolipin (CL) from *E. coli*, bovine brain L-α-phosphatidyl-L-serine (PS), EPR spin-labeled probe 5-doxylstearic acid (5-DSA) and trypan-blue dye were purchased from Sigma Chemical Co. (St. Louis, MO, USA). 3-*sn*-phosphatidic acid sodium salt from egg yolk lecithin (PA) was purchased from Merck KGaA (Darmstadt, Germany). Standard molecular weight proteins for the native PAGE were purchased from Sigma Chemical Co. (St. Louis, MO, USA). Fetal bovine serum was purchased from Thermo Fisher Scientific (Shanghai, China). RPMI-1640 medium was purchased from Sigma-Aldrich (Darmstadt, Germany). Normal human peripheral blood lymphocytes and lymphoblasts derived from an acute human T cells leukemia (Jurkat Clone E6-1 ATCC # TIB-152) were purchased from the Shanghai Biotechnology Conservation Center (Shanghai, China). All other reagents used in this study were purchased from Sigma Chemical Co. (St. Louis, MO, USA). Melittin was purified from the trace PLA_2_ contamination and other organic contaminants by cation exchange HPLC on a SCX 83-C-13-ET1 Hydropore column (Rainin Instrument, Woburn, MA, USA) as previously described [42]. Phospholipids were purified from residual contaminants on silica columns.

### 4.2. Melittin Cytotoxicity Test

Cytotoxicity of melittin was estimated by trypan blue uptake of treated and control cell samples. Lymphocytes from healthy human blood were separated by density-gradient centrifugation on lymphocyte separation medium. Jurkat cells and lymphoblasts derived from an acute human T cell leukemia were incubated in RPMI-1640 medium containing 10% fetal bovine serum and 5 mg/mL gentamicin at 37 °C in a 5% CO_2_ humidified atmosphere. Each of the cell samples was washed three times in RPMI-1640 medium and then adjusted to 10^7^ cells per mL. Healthy lymphocytes or Jurkat cells (0.5 mL) were incubated with defined concentrations of melittin for 30 min at 37 °C. The melittin-treated cell samples were centrifuged and resuspended in fresh medium, after which the cell viability was determined by trypan-blue assay. Cells incubated in the absence of melittin served as controls. Percent cytotoxicity was calculated according to the formula: Cytotoxicity, % = (number of viable cells in a test sample/number of viable cells in a control sample) × 100.

Each data point reported is the mean of three experiments ± the standard deviation. The standard deviation was always within ±5.5% of the means.

### 4.3. Effect of Melittin on Mitochondrial Respiration

Lymphocytes from healthy human blood and Jurkat cells were adjusted to aliquots of 10^7^ cells per ml as described above. Healthy human lymphocytes (HHL) and Jurkat cells were incubated in RPMI-1640 medium (0.3 M mannitol, 10 mM MOPS, 1 mM EDTA and 0.1% BSA at pH 7.4) with 10^−5^ M melittin for 30 min at 37 °C. Control cell samples were incubated in RPMI-1640 medium for 30 min at 37 °C in the absence of melittin. After incubation, the cells were pelleted by centrifugation at 370× *g* for 10 min, the supernatant was decanted, and the cells were resuspended in RPMI-1640 medium and washed twice. Mitochondria were isolated from the melittin-treated and untreated (control) cells by using sequential centrifugation steps as previously described [30]. The state 3 respiration of mitochondria in 0.3 M mannitol, 10 mM MOPS, 1 mM EDTA and 0.1% BSA at pH 7.4 was initiated by the addition of 1.5 mM succinate, 2 mM ADP, and Pi. Oxygen consumption was measured by using the Clark electrode. The mitochondrial protein concentration was 0.6 mg/mL. The respiratory control index (RCI) of mitochondria was estimated as the ratio of the O_2_ consumption in state 3 divided by the O_2_ consumption in state 4 when all ADP has been converted to ATP. In another experimental set, mitochondria isolated from HHL and Jurkat cells, and resuspended in 0.3 M mannitol, 10 mM MOPS, 1 mM EDTA and 0.1% BSA at pH 7.4, were directly treated with 10^−5^ M melittin for 30 min at 37 °C; then, mitochondria were sedimented at 10,000× *g*, the supernatant was decanted, and mitochondria were resuspended in the same medium. The RCI of mitochondria isolated from HHL and Jurkat cells, and treated directly with melittin, and the RCI of mitochondria not treated with melittin was estimated as described above. Each data point is the mean of three experiments ± the standard deviation. The standard deviation was always within ±7.1% of the means.

### 4.4. Interaction of Melittin with Model Membranes Using the Spin-Label EPR

To study the effects of melittin on the molecular mobility and orientation of the long molecular axis of phospholipids in model lipid membranes, EPR spectra of the spin probe 5-DSA in oriented multibilayer lipid films were recorded at 37 °C with a Varian E-4 spectrometer equipped with a temperature control device (Varian Inc., Palo Alto, CA, USA) at modulation amplitudes not exceeding 2 × 10^−4^ T and with a resonator input power not exceeding 20 mW. Oriented multibilayer lipid films were prepared by squeezing large unilamellar liposomes between two glass plates at a final phospholipid concentration 50 mM as previously described [25]. Large unilamellar liposomes were prepared by the ether evaporation method as previously described [43]. Oriented multibilayer lipid films were made of: (1) pure PC, (2) PC + 20 mol% PS, (3) PC + 20 mol% PA and (4) PC + 20 mol% CL. The orientation of multibilayer lipid films in the applied magnetic field was done with the resonator accessory. The analysis of the EPR spectra was done in terms of the B/C ratio [44] and the *S* parameter [45]. *B* is defined as the intensity of the low-field component, while *C* is defined as the intensity of the central component of EPR spectra taken with respect to the magnetic field perpendicular to the bilayer normal. The formula used to calculate the *S* parameter was previously published [45]. Each sample for the EPR assay was prepared and tested in triplicates. Experimental data points are the means derived from the triplicate measurements and calculations of the B/C ratio and *S* parameter. The standard deviation was always within ±3.5% of the means.

### 4.5. Assay on Melittin Association with Phospholipids in Buffers 

To measure the stable association of CL, PA, PS or PC with melittin, phospholipids and melittin were incubated in an assay buffer containing 30% ^2^H_2_O and 70% ^1^H_2_O, 10 mM Tris-HCl, pH 7.5, 0.5 mM EDTA and 1% Triton X-100. Phospholipids were added to the assay buffer dissolved in chloroform/methanol (1:1 by volume), whereas melittin was added to the assay buffer dissolved in water (30% ^2^H_2_O and 70% ^1^H_2_O). Samples of various concentrations of melittin and phospholipids (CL, PA, PS or PA) in the assay buffer were prepared and a stable association of melittin with individual phospholipids was analyzed by ^31^P NMR as previously described [28]. The ^31^P NMR spectra of phospholipids in the melittin–phospholipid complexes were recorded at 37 °C by employing a Bruker AM-300 spectrometer equipped with a temperature control device (Denver, CO, USA). The operating frequency of 121.5 MHz was achieved with the Bruker 10 mm multinuclear probe-head tuned for ^31^P using the spin frequency at 17 cycles per s. To adjust field homogeneity, the proton-free induction decay from the sample solutions was used. Continuous irradiation with the 90° pulse set at a pulse width of 13 μs, a power of 20 kHz, and a delay between pulses of 8 s was used for proton broad-band decoupling. To enhance the signal to noise ratio, a 2 Hz Lorentzian line-broadening function was employed to the total free-induction decays. Each sample in this study was prepared in triplicates. The integral intensities of ^31^P NMR signals were measured three times for each sample. The variation between measurements was less than 4%.

### 4.6. Melittin Oligomerization Test

For testing the effects of various phospholipids (CL, PS, PA or PC) on the oligomerization of melittin, 4.5 × 10^−6^ M CL was incubated with 1.5 × 10^−6^ M melittin in a buffer containing 10 mM Tris-HCl, pH 7.5, 0.5 mM EDTA and 1% Triton X-100 for one hour at 37 °C. The other phospholipids, namely PS, PA or PA, were separately incubated at 2.8 × 10^−6^ M with 2.8 × 10^−6^ M melittin in the same buffer for one hour at 37 °C. The weight of the melittin–phospholipid complexes was estimated by native PAGE as previously described [30].

### 4.7. Molecular Docking of Melittin with Phospholipids 

The polar head groups of PC, PS, PA and CL were docked with melittin (PDB code 2MLT) by using the AutoDockVina Version 4.2 program according to the protocol previously published [46]. The PDB coordinates of PC were extracted from the structure of PITP complexed to DOPC (PDB code 1T27). The PDB coordinates of PS were extracted from the crystal structure of Tim-4 bound to PS (PDB code 3BIB). The PDB coordinates of PA were extracted from the structure of the yeast cytochrome bc1 complex with a hydroxyquinone anion Qo site inhibitor bound to PA (PDB code 1P84). The PDB coordinates of CL were extracted from the crystal structure of bovine heart oxidoreductase bound to CL (PDB code 1V54). The virtual molecules of phospholipids were edited to remove the alkyl chains using the Avogadro program as previously published [47] and the overall charges were checked and energy-minimized using the AutoDock Vina Version 4.2 program. A grid box was set up with the following dimensions: center of x = 41.676; center of y = 2.983; center of z = 17.604; length of x = 36 Å; length of y = 28 Å; and length of z = 18 Å. The grid box was large enough to cover the entire surface of the melittin and a “ligand”, namely the polar head of PC, PS, PA or CL, for each molecular docking simulation. The setting for exhaustiveness was set up as 16, which provided consistent results in at least three sets of dockings for each ligand and melittin pair in this study. Phospholipid polar heads contained rotatable bonds, whereas melittin was kept as a rigid molecule for each run. The AutoDock runs were conducted in a hydrated system at pH 7.4 to allow for electrostatic interactions. Following each AutoDock run, the nine docked conformations were analyzed for ionic, ion–polar, and hydrogen bonds, and for non-polar interactions between the phospholipid polar head groups and charged, polar and non-polar amino acid residues of melittin by using the Python Molecular Viewer (MGL Tools, The Scripps Research Institute).

### 4.8. Statistics

The data points in this study are expressed as means ± standard deviations from three independent experiments. Data were analyzed by Student’s *t*-test (two-tailed) for single comparisons. Multiple group comparisons were conducted by performing one-way ANOVA followed by Bonferroni-corrected Tukey’s test. *p* values less than 0.05 were considered statistically significant.

## 5. Conclusions

The results of this study show that melittin is more cytotoxic to human lymphoblasts derived from acute T cell leukemia than to healthy human lymphocytes, an observation which agrees with our novel finding on the stronger attraction of melittin to acidic phospholipids, namely PA and PS on model membranes mimicking plasma membranes of cancer cells compared to neutral PC on model membranes mimicking plasma membranes of healthy cells. This study on model membranes suggests that melittin penetrates the plasma membrane of lymphoblastic leukemia cells with a higher proficiency than that of plasma membranes of healthy lymphocytes. It should be noted, however, that the ability of melittin to form large amounts of non-bilayer structures in the CL-containing model membrane, as shown for the first time in this study, implies that once in the cytosol, melittin diminishes the respiratory activity of the IMM in lymphoblastic leukemia cells and in healthy lymphocytes with equal efficiency. It is suggested that the melittin-induced inhibition of the functioning of IMM, but not the disintegration of the plasma membrane, is the major contributor to the cytotoxicity of melittin. It is evidently the higher concentration of melittin in the cytosol of cancer cells, caused by the higher rate of melittin penetration through plasma membranes of cancer cells, which can be held responsible for the higher cytotoxicity of melittin in cancer cells. Based on the strong ability of melittin to bind to CL and form non-bilayer structures in CL-containing membranes, it is suggested that melittin at a very low concentration may act similarly to SS-tetrapeptides in the rejuvenation of mitochondrial bioenergetics and in the mitigation or reversal of pathophysiological developments related to aging and non-cancerous diseases. The novel findings of this paper warrant further in vitro and in vivo studies on how the melittin-induced non-bilayer structures in the IMM influence mitochondrial bioenergetics to encourage or discourage the potential exploration of melittin use not only as an anti-cancer drug but also as a novel potential pharmaceutical agent to treat pathologies of mitochondrial bioenergetics not related to oncological conditions.

## Figures and Tables

**Figure 1 ijms-22-11122-f001:**
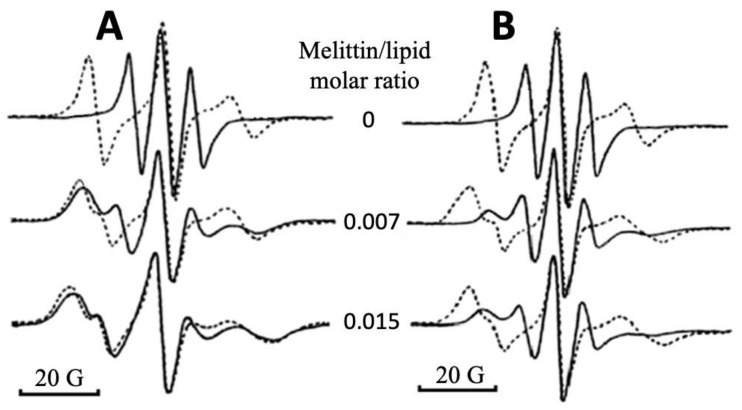
EPR spectra of 5-doxylstearic acid (5-DSA) in oriented multibilayer lipid films of PC + 20 mol% CL (**A**) and PC + 20 mol% PS (**B**) at the applied magnetic field parallel (broken line) and perpendicular (solid line) to the bilayer normal. Molar ratio of 5-DSA to lipid: 1:110.

**Figure 2 ijms-22-11122-f002:**
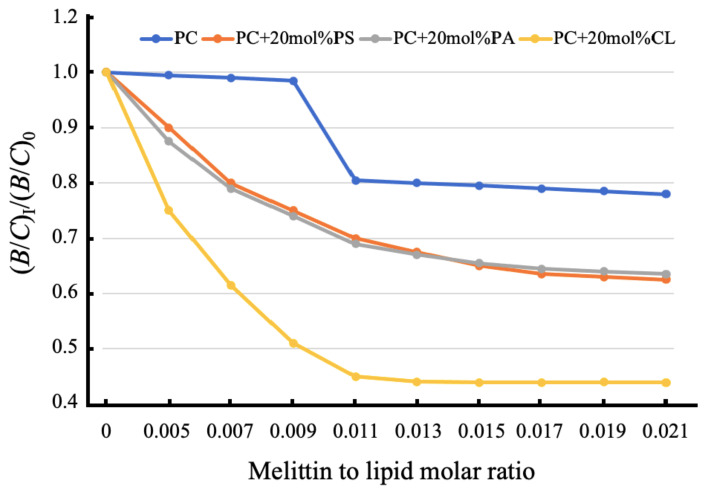
The B/C ratio of the EPR spectra of 5-DSA in oriented lipid membrane films of defined phospholipid compositions as a function of melittin concentration. Values for (B/C)_0_ represent mean B/C values from membranes without melittin and values for (B/C)_I_ represent mean B/C values from membranes treated at different melittin to lipid molar ratios. Each data point is the mean of three experiments. The standard deviation was always within ±3.5% of the means.

**Figure 3 ijms-22-11122-f003:**
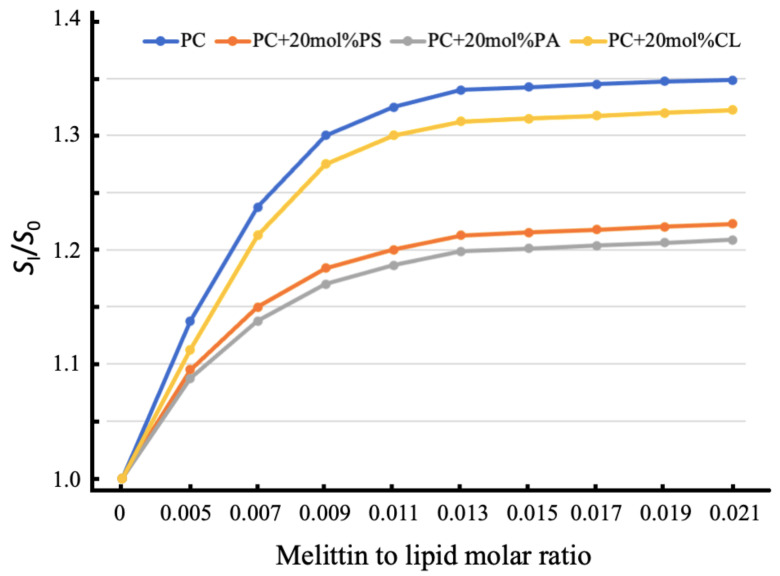
The *S* parameter of the EPR spectra of 5-DSA in oriented lipid membrane films of defined phospholipid compositions as a function of melittin concentration. Values for *S*_0_ represent mean *S* values from membranes without melittin and values for *S*_I_ represent mean *S* values from membranes treated at different melittin to lipid molar ratios. Each data point is the mean of three experiments. The standard deviation was always within ±3.5% of the means.

**Figure 4 ijms-22-11122-f004:**
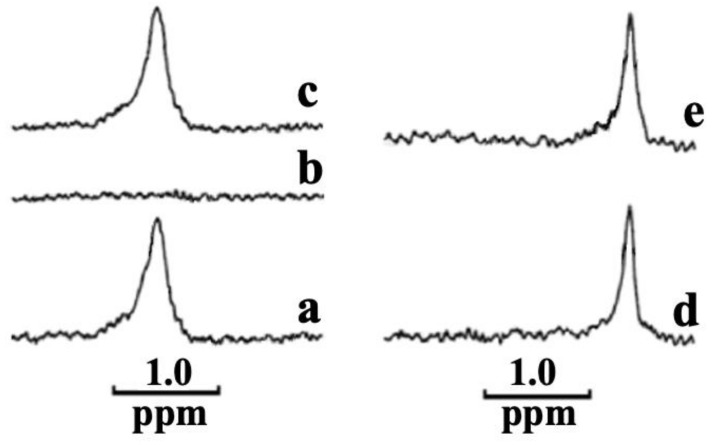
^31^P NMR spectra of CL (**a**–**c**) and PS (**d**,**e**) in buffers containing 10 mM Tris-HCl, pH 7.5, 0.5 mM EDTA, 1% Triton X-100. (**a**) ^31^P NMR spectrum of 4.5 × 10^−6^ M CL in the buffer. (**b**) ^31^P NMR spectrum of 4.5 × 10^−6^ M CL taken after one hour of incubation in the buffer with 1.5 × 10^−6^ M melittin. (**c**) ^31^P NMR spectrum of the sample (**b**) after the addition of 1% SDS to the buffer. (**d**) ^31^P NMR spectrum of 8.4 × 10^−6^ M PS in the buffer. (**e**) ^31^P NMR spectrum of 8.4 × 10^−6^ M PS taken after one hour of incubation with 2.8 × 10^−6^ M melittin.

**Figure 5 ijms-22-11122-f005:**
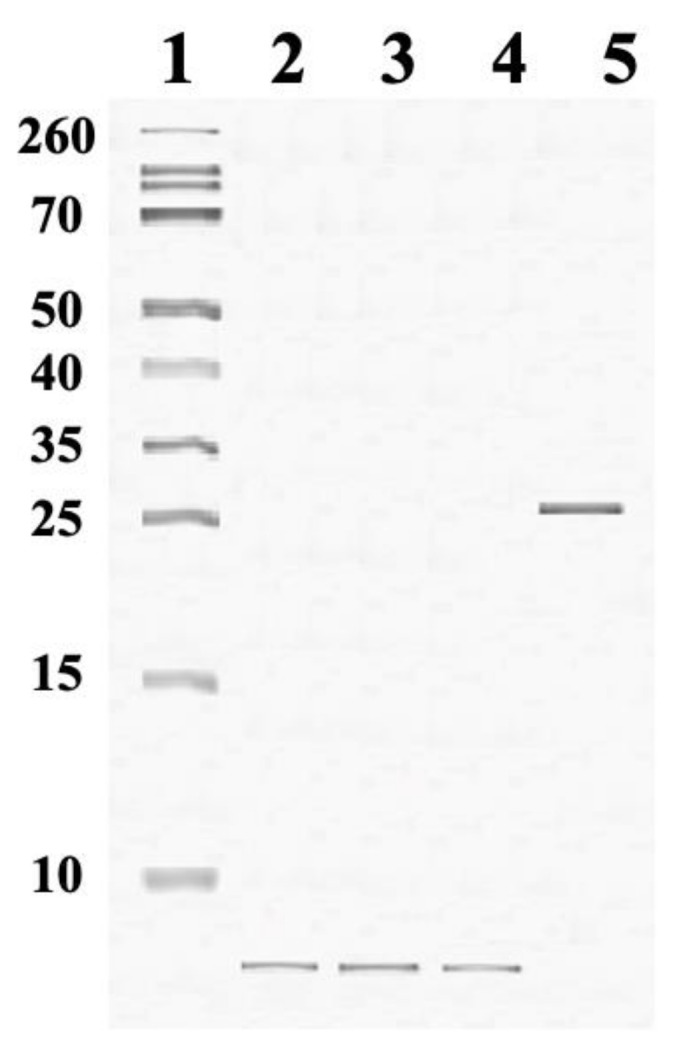
Native PAGE, performed according to [30], including standard molecular weight proteins from Sigma Chemical Co., St. Louis, MO, USA (lane **1**); 2.8 × 10^−6^ M melittin incubated with 8.4 × 10^−6^ M of either PS (lane **2**), PA (lane **3**) or PC (lane **4**); and 1.5 × 10^−6^ M melittin incubated with 4.5 × 10^−6^ M CL (lane **5**) in 10 mM Tris-HCl, pH 7.5, 0.5 mM EDTA, 1% Triton X-100 buffer for one hour at 37 °C.

**Figure 6 ijms-22-11122-f006:**
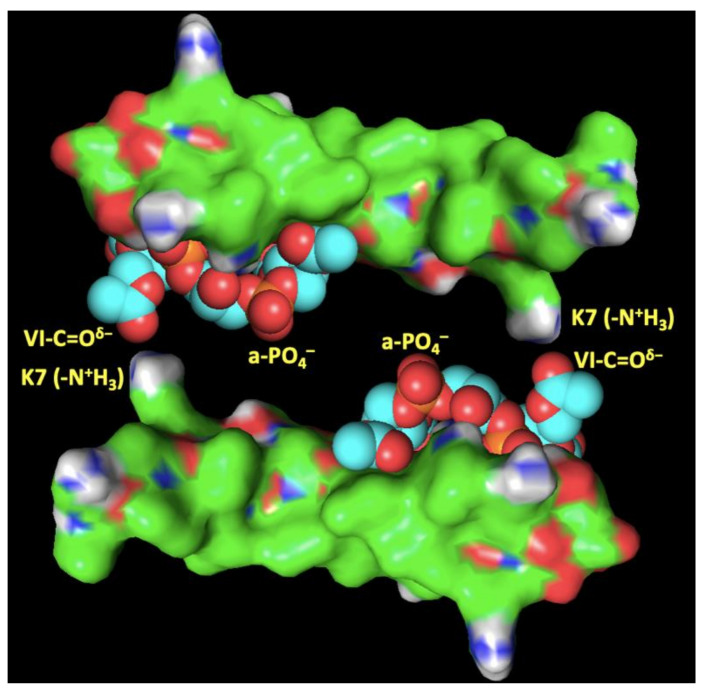
A hypothetical model of melittin dimerization mediated by the CL polar head, which may take place in the inner mitochondrial membrane. The binding site and conformation of the CL polar head on the molecular surface of melittin is predicted by the AutoDock program. Melittin is rendered as a molecular surface representation and the CL polar head is rendered as a sphere representation. It is suggested that this mode of melittin dimerization is supported by the short-distance ion–polar bond between the IV-C = O^δ−^ group of the CL polar head and the -N^+^H_3_ group of K7, and by a long-distance ionic bond between the a-PO_4_^−^ group of the CL polar head and the -N^+^H_3_ group of K7, as shown in the figure. The designation of CL polar head groups is shown in Appendix A.

**Table 1 ijms-22-11122-t001:** Cytotoxicity of melittin on healthy human lymphocytes and Jurkat cells derived from acute human T cell leukemia. The reported data are fractions of viable cells after the treatments with the defined concentrations of melittin.

Melittin Concentration (M)	Healthy Human Lymphocytes, %	Jurkat Cells, %
0	100 *	100
10^−10^	98.1 ± 3.0	95.9 ± 3.5
10^−9^	98.0 ± 5.0	85.2 ± 3.4
10^−8^	98.0 ± 3.3	71.1 ± 2.5
10^−7^	97.8 ± 3.6	53.0 ± 2.6
10^−6^	96.7 ± 3.1	39.3 ± 2.0
10^−5^	73.8 ± 4.0	25.2 ± 1.5
10^−4^	72.2 ± 3.4	21.5 ± 1.1

* Percent of viable cells is based on control cell samples not treated with melittin. The data are based on the means of three preparations ± the standard deviation. The standard deviation was always within ±5.5% of the means.

**Table 2 ijms-22-11122-t002:** Oxygen consumption in states 3 and 4 and respiratory control index of mitochondria isolated from healthy human lymphocytes (HHL) and Jurkat cells from acute human T cell leukemia after the cells were treated or untreated with melittin and of mitochondria from HHL and Jurkat cells which were directly treated or untreated with melittin. Each data point is the mean of three experiments ± the standard deviation. The standard deviation was always within ±7.1% of the means.

			HHL	Jurkat
**Cells**	**melittin-untreated**	State 3 *	81.0 ± 5.6	81.4 ± 5.7
		State 4 *	12.4 ± 0.8	12.0 ± 0.7
		RCI	6.5 ± 0.4 100%	6.8 ± 0.4 100%
	**melittin-treated**	State 3 *	70.2 ± 5.0	21.6 ± 1.4
		State 4 *	15.6 ± 1.0	12.7 ± 0.9
		RCI	4.5 ± 0.3 69.2%	1.7 ± 0.1 25.0%
**Mitochondria**	**melittin-untreated**	State 3 *	63.8 ± 4.3	64.5 ± 4.3
		State 4 *	12.5 ± 0.7	12.4 ± 0.8
		RCI	5.1 ± 0.2 100%	5.2 ± 0.3 100%
	**melittin-treated**	State 3 *	3.9 ± 0.27	4.0 ± 0.28
		State 4 *	4.1 ± 0.28	4.25 ± 0.3
		RCI	0.95 ± 0.07 18.6%	0.94 ± 0.06 17.9%

* Oxygen consumption in states 3 and 4 is given in µM min^−1^ mg^−1^ mitochondrial protein. The O_2_ consumption in state 3 was measured after the addition of 1.5 mM succinate, 2 mM ADP and Pi. The O_2_ consumption in state 4 was measured after all ADP has been converted to ATP.

**Table 3 ijms-22-11122-t003:** The binding affinity values for the polar heads of CL, PS, PA and PC for the best nine binding sites on the molecular surface of melittin predicted by the AutoDock program. A binding affinity value represents the amount of energy released in the exothermic process when intermolecular bonds are made between phospholipid polar heads and the molecular surface of melittin at a specific binding site.

	Binding Affinities (kcal/mol)			
Binding Sites Number	CL Polar Head	PS Polar Head	PA Polar Head	PC Polar
1	−4.1	−3.8	−3.5	−3.3
2	−4.0	−3.7	−3.4	−3.2
3	−4.0	−3.6	−3.4	−3.2
4	−3.9	−3.6	−3.3	−3.2
5	−3.9	−3.6	−3.3	−3.1
6	−3.9	−3.6	−3.3	−3.1
7	−3.9	−3.6	−3.2	−3.1
8	−3.9	−3.5	−3.2	−3.1
9	−3.9	−3.5	−3.2	−3.1

## Data Availability

All important data is included in the manuscript.

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
