# Peer review of "Bee Venom Melittin Disintegrates the Respiration of Mitochondria in Healthy Cells and Lymphoblasts, and Induces the Formation of Non-Bilayer Structures in Model Inner Mitochondrial Membranes"

_ijms, 2021, doi:10.3390/ijms222011122_

Round 1

Reviewer 1 Report

This is a manuscript reporting on ‘Bee Venom Melittin Disintegrates Respiration of Mitochondria in Healthy and Leukemia Lymphocytes and Induces Formation of Non-bilayer Structures in Model Inner Mitochondrial Membranes’’.  The authors report on greater cytotoxic effects of melittin on Jurkat T-ALL cell line compared to healthy lymphocytes, but they report no difference in O2 consumption and RCI values between Jurkat and healthy lymphocytes when mitochondria were directly treated with melittin. They also performed experiments on model membranes showing properties of melittin to form non-bilayer structures in CL-containing membranes mimicking IMM. They report that melittin-mediated cytotoxicity is achieved due to the higher membrane permeability of cancer cells, accumulation of melittin in cytosol and consequently disintegration of IMM.

Broad comments: This is relatively readable manuscript reporting on mechanisms of melittin-mediated cytotoxicity.  Although the scientific background is not so bad, the experimental design should be vastly improved.

Specific comments:

  1. In cytotoxicity and O2 consumption experiments authors compare healthy human lymphocytes to Jurkat cell line (T-ALL) but the normal counterpart of Jurkat cells would be T-cells and not all lymphocytes. If authors want to compare healthy human lymphocytes to ALL cells, to eliminate properties specific only to Jurkat cells or to T-cell lineage, at least one more cell line derived from B-ALL should be investigated (eg. REH, NALM-6, SUP-B15). If authors wish to elaborate on T-specific effects of melittin, control cells should be T lymphocytes and experimental group should have one more T-ALL derived cell line (eg. MOLT-6, CCRF-CEM)
  2. Melittin-mediated mechanisms on Jurkat cells are further investigated on model membranes and authors draw conclusions from those models on the mechanism of melittin on Jurkat cells. The authors cite ref 21 and 21 where no phospholipid composition of ALL is reported. Authors should in detail elaborate why were exactly 20 mol% PS and PA chosen to represent cells of acute lymphoblastic leukemia especially when it was shown that concentration of PS in ALL is decreased (PMID: 18839072)? Additional experiments on the concentration of phospholipids in ALL cell lines in comparison to human healthy lymphocytes should be conducted before generating optimal cancer model membrane system.
  3. The authors show the equal RCI values of HHL and Jurkat cells and discuss that mitochondrial bioenergetics of cancer cells is not much different from healthy cells. There are numerous reports on different mitochondrial activity in cancer cells compared to healthy cells (PMID: 33409153, PMID: 32764295, PMID: 32168755, PMID: 34561557). The authors should discuss in detail their results and results on additional cell lines to be performed in the light of these publications.
  4. In the title, abstract and main text authors state leukemia lymphocytes which is false and should be replaced with term lymphoblasts or acute lymphoblastic leukemia cells. Further, Jurkat cells shoud be more defines in M&M section – the cells in this study are supposedly Jurkat, Clone E6-1 (in opposition to eg BCL2 Jurkat which are T lymphocytes)
  5. In introduction section, more recent publications reporting on melittin parmacological effects should be cited, eg. there is an interesting report by Duffy et al. on mechanism of melittin on breast cancer (PMID: 34465853)
  6. The subtitles in the results section should be added.

Author Response

Comments and Suggestions for Authors

This is a manuscript reporting on ‘Bee Venom Melittin Disintegrates Respiration of Mitochondria in Healthy and Leukemia Lymphocytes and Induces Formation of Non-bilayer Structures in Model Inner Mitochondrial Membranes’’.  The authors report on greater cytotoxic effects of melittin on Jurkat T-ALL cell line compared to healthy lymphocytes, but they report no difference in O2 consumption and RCI values between Jurkat and healthy lymphocytes when mitochondria were directly treated with melittin. They also performed experiments on model membranes showing properties of melittin to form non-bilayer structures in CL-containing membranes mimicking IMM. They report that melittin-mediated cytotoxicity is achieved due to the higher membrane permeability of cancer cells, accumulation of melittin in cytosol and consequently disintegration of IMM.

Broad comments: This is relatively readable manuscript reporting on mechanisms of melittin-mediated cytotoxicity.  Although the scientific background is not so bad, the experimental design should be vastly improved.

Specific comments:

Authors responses to the Reviewer 1 comments are given in the blue letter sentences below:

  1. In cytotoxicity and O2 consumption experiments authors compare healthy human lymphocytes to Jurkat cell line (T-ALL) but the normal counterpart of Jurkat cells would be T-cells and not all lymphocytes. If authors want to compare healthy human lymphocytes to ALL cells, to eliminate properties specific only to Jurkat cells or to T-cell lineage, at least one more cell line derived from B-ALL should be investigated (eg. REH, NALM-6, SUP-B15). If authors wish to elaborate on T-specific effects of melittin, control cells should be T lymphocytes and experimental group should have one more T-ALL derived cell line (eg. MOLT-6, CCRF-CEM)

The reviewer raised a very good point regarding the samples of cancer and healthy cells as in our ms we failed to explain our choices for cancer and healthy cells. Cytolytic activity of melittin and its effects on cell respiration of cancer and healthy white blood cells have been reported earlier by Dr. Mirtalipov with the particular focus on two types of cancer cells: T lymphoblastic and B lymphoblastic leukemia cells, and three types of healthy cells: normal peripheral blood lymphocytes, normal T cells, and normal B cells. Although melittin’s effects were stronger on samples of cancer cells, as compared to samples of healthy cells, there was no difference reported in the effects of melittin between the two types of cancer blood cells and between the three types of healthy blood cells. This was an expected observation as Dr. Mirtalipov previously reported results of his TLC assay on lipid composition of plasma membranes of cancer and healthy blood white cells, which showed insignificant difference between T and B lymphoblasts and normal T and B cells (although an increase in acidic phospholipids was observed in lymphoblasts as compared to normal lymphocytes). For this reason, we used only one type of cancer blood cells (T-ALL) and one type of healthy blood cells (normal peripheral blood lymphocytes) in our study. Now, we explain this point in the revised manuscript (lines 117-133 in the revised manuscript).

  1. Melittin-mediated mechanisms on Jurkat cells are further investigated on model membranes and authors draw conclusions from those models on the mechanism of melittin on Jurkat cells. The authors cite ref 21 and 21 where no phospholipid composition of ALL is reported. Authors should in detail elaborate why were exactly 20 mol% PS and PA chosen to represent cells of acute lymphoblastic leukemia especially when it was shown that concentration of PS in ALL is decreased (PMID: 18839072)? Additional experiments on the concentration of phospholipids in ALL cell lines in comparison to human healthy lymphocytes should be conducted before generating optimal cancer model membrane system.

This is a valid comment by the reviewer. References 21 and 22 report increased exposure of anionic phospholipids and PS as a biomarker on surface of cancer cells, but not on surface of human ALL. We have now included reference 47, which reports the phospholipid composition of the outer leaflet of human ALL and healthy lymphocytes and the overall phospholipid composition of IMM in ALL and healthy lymphocytes. We have now given a detailed explanation for the phospholipid compositions in model membranes for the outer surface of ALL and healthy lymphocytes and for overall phospholipid composition of IMM (lines 216-246 in the revised manuscript).

As to the report PMID: 18839072, we would like to note that in the cited work 31P-NMR was used to record signals of phospholipid extracts. This does not necessarily give information on the phospholipid composition of the outer leaflet of plasma membrane. There are many works reporting increase in anionic phospholipids of the membrane’s outer leaflet in cancer cells; which, however, does not always mean overall increase in anionic phospholipids. Also, it is possible that in PMID: 18839072 SDS was not used to remove the annular phospholipids. Concentration of the annular phospholipids tend to increase in cancer cells, particularly anionic phospholipids exposed on the outer leaflet.

  1. The authors show the equal RCI values of HHL and Jurkat cells and discuss that mitochondrial bioenergetics of cancer cells is not much different from healthy cells. There are numerous reports on different mitochondrial activity in cancer cells compared to healthy cells (PMID: 33409153, PMID: 32764295, PMID: 32168755, PMID: 34561557). The authors should discuss in detail their results and results on additional cell lines to be performed in the light of these publications.

We are thankful to the reviewer for this comment. However, the reports suggested by the reviewer, and other reports on different mitochondrial activity in cancer cells compared to healthy cells relate to cases involving different variables including different types of cancer and group of patients (sex, age, progression of disease, type of treatment, associated side effects etc.), and different methods of investigation which has little relevance to the scope of our investigation. In our paper we present a study in which we report the same rate of respiration and coupling in lymphoblasts and healthy lymphocytes (just a little more efficient in lymphoblasts) and we see the same level of degradation of respiration and coupling when the IMM in cancer and healthy cells are directly treated with melittin. However, when intact cells are treated with melittin, respiration and coupling of healthy cells is more viable than those in cancer cells. We relate these observations to the lipid packing and dynamics in melittin-treated model membranes mimicking the outer surface of plasma membrane and IMM in cancer and healthy cells; and we conclude that melittin’s greater cytotoxic activity against cancer cells relates to the higher permeability of the cancer-cells plasma membranes. As a bonus of our study, we discovered that melittin triggers the formation of non-bilayer structures and closely interacts with the main non-bilayer lipid, cardiolipin. Hence, we propose that melittin should be tried as a pharmaceutical agent in rejuvenation of mitochondrial bioenergetics, a proposal similar to the one in our previous study discovering that cardiotoxin from cobra venom facilitates ATP synthesis in intact mitochondria via enhancing non-bilayer structures (refs. 2, 29, 36).         

  1. In the title, abstract and main text authors state leukemia lymphocytes which is false and should be replaced with term lymphoblasts or acute lymphoblastic leukemia cells. Further, Jurkat cells shoud be more defines in M&M section – the cells in this study are supposedly Jurkat, Clone E6-1 (in opposition to eg BCL2 Jurkat which are T lymphocytes)

We thankfully agree with the reviewer. The Jurkat cell line we used in our study is ATCC # TIB-152 which we now defined in M&M on line 686 in the revised manuscript. We also replaced the term leukemia lymphocytes by lymphoblasts or by acute lymphoblastic leukemia cells in the title, abstract and the main text (shaded in yellow color).  

  1. In introduction section, more recent publications reporting on melittin parmacological effects should be cited, eg. there is an interesting report by Duffy et al. on mechanism of melittin on breast cancer (PMID: 34465853)

This report by Duffy suggested by the reviewer is an interesting article which offers a mechanism of melittin’s anti-cancer selectivity mostly linked to the suppression of growth factor receptors and this article mentions binding of melittin to the negatively charged plasma membrane. Now, we cite this article in the Introduction along with other recent publications (shaded in yellow color in References). 

  1. The subtitles in the results section should be added.

As per the reviewer’s request, we have added subtitles in the results section.

Reviewer 2 Report

Overall, the study is well-designed and reported, and I believe some of the observations obtained are quite novel and will add significant knowledge to the community. Overall, it is a sound body of work, the paper is well-written but I have few comments which will help strengthen the manuscript.

Major comments:

  1. The authors indirectly measured the effect of melittin on mitochondrial membrane potential of healthy and leukemic cells by observing its effects on model membranes such as PC and PC + 20 mol% PS or PA, respectively. However, a more biologically relevant and robust assay would be to analyze the effect of melittin on the mitochondrial activity using dyes such as tetramethylrhodamine-ethyl ester (TMRE) to assess mitochondrial membrane potential and mitoSOX redRdye for mitochondrial ROS detection in the cells. This will give a more broader and comprehensive picture.

Minor comments:

  1. Data presented in Table 1 must be presented in the form of a concentration response curve or a sigmoid curve. This will also help in calculating the IC50 of melittin on healthy and leukemic cells.

Author Response

Comments and Suggestions for Authors

Overall, the study is well-designed and reported, and I believe some of the observations obtained are quite novel and will add significant knowledge to the community. Overall, it is a sound body of work, the paper is well-written but I have few comments which will help strengthen the manuscript.

Authors responses to the Reviewer 1comments are given in the blue letter sentences below:

Major comments:

  1. The authors indirectly measured the effect of melittin on mitochondrial membrane potential of healthy and leukemic cells by observing its effects on model membranes such as PC and PC + 20 mol% PS or PA, respectively. However, a more biologically relevant and robust assay would be to analyze the effect of melittin on the mitochondrial activity using dyes such as tetramethylrhodamine-ethyl ester (TMRE) to assess mitochondrial membrane potential and mitoSOX redRdye for mitochondrial ROS detection in the cells. This will give a more broader and comprehensive picture.

We thank the reviewer for this comment. However, we must emphasize that we did not measure the effects of melittin on the mitochondrial membrane potentials. We used model membranes (oriented multibilayer films), which mimicked phospholipid composition of healthy cells and lymphoblast and the IMM, to measure the effect of melittin on phospholipids’ packing and rotational movement by the EPR of spin probe. However, in our future studies we will consider using tetramethylrhodamine-ethyl ester (TMRE) and mitoSOX redRdy for assessing the effects of melittin on mitochondrial membrane potential and on changes in mitochondrial ROS to obtain biologically more relevant and robust parameters on changes in mitochondrial bioenergetics.

Minor comments:

  1. Data presented in Table 1 must be presented in the form of a concentration response curve or a sigmoid curve. This will also help in calculating the IC50 of melittin on healthy and leukemic cells.

            We appreciate this comment by the reviewer, however, since the results of melittin             cytotoxicity test presented in Table 1 showed that melittin was able to kill only a fraction in         populations of healthy cells and lymphoblasts, we thought that the curve of cell viability to         concentration responses cannot help in calculating IC50.

            Overall, we are deeply grateful to this reviewer for his/her careful consideration and review of our     manuscript and for his/her appreciation of the results of our work.

Round 2

Reviewer 1 Report

I have carefully read the revised version of the manuscript ''Bee Venom Melittin Disintegrates Respiration of Mitochondria in Healthy and Leukemia Lymphocytes and Induces Formation of Non-bilayer Structures in Model Inner Mitochondrial Membranes''. The authors have addressed all of the issued raised and no further revisions are needed. 

Reviewer 2 Report

Accept in the present form

This manuscript is a resubmission of an earlier submission. The following is a list of the peer review reports and author responses from that submission.